# Impact of Early Childhood Malnutrition on Cardiometabolic Risk Factors in Young Adults from Marginalized Areas of Chiapas, Mexico

**DOI:** 10.3390/nu17020254

**Published:** 2025-01-11

**Authors:** Fátima Higuera-Domínguez, Héctor Ochoa-Díaz-López, César Antonio Irecta-Nájera, Pilar E. Núñez-Ortega, Itandehui Castro-Quezada, Rosario García-Miranda, Roberto Solís-Hernández, Esmeralda García-Parra, María Dolores Ruiz-López

**Affiliations:** 1Health Department, El Colegio de la Frontera Sur, Carr. A Reforma Km. 15.5 s/n, RA. Guineo 2da. Sección, Villahermosa 86280, Mexico; fatima.higuera@estudianteposgrado.ecosur.mx (F.H.-D.);; 2Faculty of Nutrition and Food Science, University of Science and Arts of Chiapas, Libramiento Norte-Poniente 1150, Col Lajas Maciel, Tuxtla Gutiérrez 29039, Mexico; 3Faculty of Human Medicine, Autonomous University of Chiapas, Calle Central-Sur s/n, Col. San Francisco, Tuxtla Gutiérrez 29000, Mexico; 4Health Department, El Colegio de la Frontera Sur, Carr. Panamericana y Periférico Sur s/n, Barrio María Auxiliadora, San Cristobal de Las Casas 29290, Mexico; 5School of Languages-Campus San Cristobal, Autonomous University of Chiapas, Javier Lopez Moreno S/N, Barrio de Fatima, San Cristobal de las Casas 29264, Mexico; 6Faculty of Pharmacy, Department of Nutrition and Food Science, University of Granada, 18071 Granada, Spain; 7Biomedical Research Center, Institute of Nutrition and Food Technology “José Mataix”, University of Granada, 18071 Granada, Spain

**Keywords:** Childhood malnutrition, cardiometabolic risk factors, obesity

## Abstract

The presence of malnutrition in early life is a determining factor in the onset of metabolic alterations and chronic diseases in adults. Therefore, the objective of this study was to determine the impact of malnutrition in early childhood with the presence of cardiometabolic risk factors in adulthood in marginalized populations from Chiapas, Mexico. The present investigation was based on a prospective cohort study that began in 2002, with young adults aged 18 to 25 years belonging to *De Los Bosques* region in Chiapas, Mexico. Sociodemographic, anthropometric, clinical and biochemical data were obtained in adulthood. Binary logistic regression models with 95% confidence intervals were fitted to assess the association between nutritional status in childhood (≤5 years of age) and cardiometabolic risk in adulthood. Individuals with overweight/obesity in childhood were more likely to have overweight/obesity (OR = 2.65, 95% CI: 1.09–6.45), high waist circumference (3.78, 95% CI: 1.55–9.24), high waist to height ratio (OR = 5.38, CI 95%: 1.60–18.10), elevated total cholesterol (OR = 3.95, 95% CI: 1.36–11.43) and metabolic syndrome (OR = 4.71, 95% CI: 1.49–14.90) in adulthood. In conclusion, malnutrition presented in early childhood increased the probability of developing cardiometabolic alterations in young adults from southern Mexico.

## 1. Introduction

Malnutrition is one of the main causes of morbidity and mortality worldwide [1]. Overweight and obesity have increased worldwide, mainly affecting low- and middle-income countries [1,2]. There are more than five million deaths each year from diseases attributable to overweight and obesity, which represents almost 9% of the lives lost annually [3]. 

On the other hand, undernutrition represents more than half of global deaths in children under five years of age [4,5]; this is a situation that represents a public health problem [6].

Malnutrition (in both forms: undernutrition and obesity) present in early childhood predisposes to the development of chronic non-communicable diseases in the different cycles of life, which represents a major health problem worldwide [7,8].

An unhealthy nutritional intake in the first years of life can cause stunting and obesity during childhood, subsequently affecting nutritional status at different stages of life, increasing the likelihood of developing diseases such as type 2 diabetes and cardiovascular diseases [7,9,10], which are the leading causes of mortality in Mexico and Chiapas [11,12].

The effects of overweight and obesity at an early age often continue into adulthood, increasing the population’s susceptibility to presenting cardiometabolic risk factors such as central adiposity, insulin resistance (IR), altered glucose metabolism, dyslipidemia and high blood pressure [9,13]. Different studies have supported the idea that overweight and obesity in childhood are consistently associated with the development of cardiometabolic risk factors and metabolic syndrome (MetS) in adults [14].

Although the prevalence of overweight and obesity in children under 5 years of age has decreased in Mexico in the last ten years, it remains as a health priority due to the long-term consequences mentioned above. National data issued by the 2021 National Health and Nutrition Survey showed a prevalence of 8.4% of overweight/obesity in children under five years of age and 74.1% in adults [15].

An increase in cardiovascular risk factors has been found in the rural population due to urbanization processes, nutrition transition and various environmental and genetic conditions [16]. In the southeast of Mexico, the abandonment of traditional agricultural activity, migration and political and social problems, such as development inequalities during the 20th century, have favored the phenomenon known as “the double burden of malnutrition” [17,18], described as the presence of both overweight/obesity and undernutrition in at least one group/region of the population [19].

Approximately 14% of the extreme poverty in Mexico is concentrated in Chiapas. Chiapas occupies the first place of marginalization as a result of the lack of access to education and health services, living in inadequate housing and low monetary income, which result in precarious opportunities that obstructs the full development of human potential, experiencing discrimination and social, political and economic exclusion [20].

In Chiapas, there are few studies investigating the association between malnutrition and long-term health. Thus, the objective of this study was to determine the impact of malnutrition in early childhood with the presence of cardiometabolic risk factors in adulthood, 18 years later after the first evaluation, in marginalized populations from the state of Chiapas, Mexico.

## 2. Materials and Methods

### 2.1. Study Area

This study was carried out in four communities; two at the municipality of Huitiupán (La Competencia and Ramos Cubilete) and two at the municipality of Simojovel (El Jardín and Rivera Domínguez), which belong to the De Los Bosques region, Chiapas, Mexico. (Figure 1).

### 2.2. Study Design and Population

The present investigation represents a cross-section of a follow-up study regarding young adults aged 18 to 25 years old, based on a prospective cohort study of children under five years started in 2002, belonging to the De Los Bosques region, Chiapas, Mexico.

The details of the cohort study have been previously published [21,22,23]: 407 children under five years of age were studied during 2002–2003, living in four communities; two at the municipality of *Huitiupán* (*La Competencia* and *Ramos Cubilete*) and two at the municipality of *Simojovel* (*El Jardín* and *Rivera Domínguez*). An intentional sampling technique was carried out, taking into account the research objectives at that time. The municipalities were selected according to their level of marginalization, ethnic condition and priority indigenous area from the Secretary of Health of Chiapas. The communities were selected based on their geographic access (two with difficult access and two near to the municipal capital), indigenous and non-indigenous population and affiliation with health services for the general population. All homes in the four selected communities were visited, obtaining the aforementioned sample. In 2004–2005 and 2010–2011, a second and third nutritional evaluation was carried out; 93% of the studied population, due to their poverty condition, received the benefits of the *Oportunidades* Program (a governmental anti-poverty program that provided poor families a monetary incentive, food supplements, health care and nutritional education) [24]. In 2004–2005 and 2010–2011, a second and third nutritional evaluation was carried out on the population belonging to the cohort, data that were not used in this research. For the present study, performed in 2020–2021, 407 young adults from the original cohort were sought, obtaining a final sample of 122 individuals (76 women and 46 men) who were interviewed, out of which 97 (79.5%) agreed to provide biological samples (blood) (Figure 2).

### 2.3. Data Collection

#### 2.3.1. Instrument

A semi-structured and pre-coded questionnaire was carried out through interviews to obtain the following data: sociodemographic characteristic, including the following: age (years); sex (female or male); marital status (single or with a partner); language (Spanish or local languages); education (<6 years or ≥6 years); reading and writing (read and write or illiterate); occupation (classified into two categories: agriculture and home, and other job); household conditions (fridge and television); non-pathological personal history (alcohol consumption and smoking); anthropometric, clinical and biochemical measurements; food intake through a single 24 h recall; physical activity.

#### 2.3.2. Anthropometric Measurement

Overweight/obesity at five or less years old was defined as a weight-for-height (z-score ≥ +1 SD) and stunting as height-for-age (z-score ≤ −2 SD), both using World Health Organization (WHO) criteria [25].

Anthropometric parameters were obtained during the interview according to the International Society for the Advancement of Kinanthropometry (ISAK) protocols and quality procedures [26]. The body weight of the young adult participants was evaluated using electronic scales calibrated with a precision of ±100 g (TANITA UM081; Tokyo, Japan). Stadiometers measured height with a precision of ±1 mm (SECA E123; Berlin, Germany). Body mass index (BMI) was calculated by dividing the weight by the square of the height (kg/m^2^). BMI was categorized using the cut-off points of the WHO [27]: normal = BMI < 25 m/kg^2^, overweight = BMI < 30 m/kg^2^; obesity = BMI ≥ 30 m/kg^2^. Waist circumference (WC) was measured with a SECA 201 measuring tape with a precision of ±1 mm (SECA GmbH & Co. KG., Hamburg, Germany) and was classified using the criteria of the International Diabetes Federation (IDF) as follows: WC ≥ 80 cm for women and WC ≥ 90 cm for men [28]. The waist to height ratio (WHtR) was determined by dividing WC (cm) by height (cm), using the cut-off point ≥ 0.5 units as a marker for cardiovascular risk [29].

#### 2.3.3. Clinical Measurement and Biochemical Parameters

Blood pressure was measured twice (in the middle and at the end of the interview) with a Citizen digital baumanometer, model 453-CH (Citizen Systems Japan Co., Ltd., model CH-453, Tokio, Japan) with a precision of ±3 mmHg. The participant was sitting in a resting position, and the measurement was performed on the dominant arm.

The blood sample was taken from the vein located in the antecubital fossa of the arm. The sample was collected in vacutainer tubes without anticoagulant after fasting for 12 h. The serum was separated by centrifugation (6000 rpm) and placed in Eppendorf vials to be transferred in the cold chain to the Health Laboratory “Molecular Epidemiology and Nutrigenomics” of El Colegio de la Frontera Sur (ECOSUR) San Cristóbal and stored at a temperature of −80 ° C (Ultra-low freezer). Mandatory hygiene and safety measures were carried out before, during and after the blood sample was collected.

The quantitative determination of serum glucose, total cholesterol, triglycerides, high-density lipoprotein cholesterol (HDL-c) and low-density lipoprotein cholesterol (LDL-c), were performed by the corresponding standard enzymatic colorimetric method for each analyte (Spinreact). Insulin analysis was performed in duplicate using the ELISA method (Invitrogen). Insulin resistance was obtained by the homeostasis model (HOMA-IR): [(fasting insulin (µU/mL) × fasting glucose (mg/dL))/22.5] [30]. Spectrometry equipment (Multiskan GO microplate spectrophotometer—Thermo Scientific, Waltham, MA, USA) was used in all analyses. The cut-off points used to categorize biochemical parameters are shown in Table 1.

#### 2.3.4. Metabolic Syndrome Classification

Metabolic syndrome classification (MetS) was determined using the IDF criteria [28]: WC for women ≥ 80 cm and men ≥ 90 cm, plus two of the following variables: basal glucose ≥ 100 mg/dL triglycerides ≥ 150 mg/dL, HDL-c ≤ 50 mg/dL for women and ≤40 mg/dL for men and systolic/diastolic pressure ≥ 130/85 mmHg.

#### 2.3.5. Covariates

##### Dietary Assessment

A single 24 h dietary recall was applied to estimate the total energy intake. The size of the portion consumed by the participant was estimated with the help of household measures. The reported food intake was converted to grams per day to obtain total energy intake (kcal/day). The Nutrient Composition Tables for Mexican Foods [32] and The Food Composition Database of the United States Department of Agriculture (USDA) [33] were used. 

##### Physical Activity Assessment

Physical activity was assessed through the Global Physical Activity Questionnaire (GPAQ) developed by the WHO [34]. The Metabolic Equivalent of Task (MET) per week (MET/week) was determined and categorized as follows: active (>600 MET/week) or sedentary (<600 MET/week).

### 2.4. Statistical Analysis

Statistical analyzes were performed using SPSS software (version 23, 2018, BM SPSS, Armonk, NT, USA). To determine whether or not the data had a normal distribution, the Kolmogorov–Smirnov test was used. Descriptive data are presented as measures of central tendency and dispersion as medians and interquartile ranges (IQRs), as well as percentages. Qualitative variables were compared by sex using Pearson’s chi squared tests, and Mann–Whitney U tests were used for qualitative variables. A *p*-value of ≤0.05 was considered statistical significance for all analyses. Intercorrelations between independent variables were assessed by Spearman’s correlation.

Binary logistic regression models were used to determine the association between malnutrition in early childhood (exposure) and the various anthropometric, biochemical and clinical variables and MetS (outcomes). Odds ratios (ORs) with 95% confidence intervals (CI) were calculated. For each outcome, crude models were tested, and models were adjusted separately for each outcome for potential confounding factors: sex (categorical variable: female and male), age (continuous variable), language (categorical variable: Spanish or Indigenous), total energy intake (continuous variable: kcal/day) and physical activity (categorical variable: active or sedentary). A *p*-value < 0.05 in the likelihood ratio test was considered statistically significant.

### 2.5. Ethical Considerations

Written consent was obtained from each participant prior to the interview. The study was approved by the Research Ethics Committee of El Colegio de la Frontera Sur in accordance with the Declaration of Helsinki (202021013/3 May 2021).

## 3. Results

No statistically significant differences in sex were found according to the sociodemographic characteristics (Table 2); the mean age of the study population was 21 ± 3 years and 51.6% were single. In total, 35.2% of the participants spoke a local language and 81.1% had more than six years of schooling. Most of the population (77.9%) was dedicated to agriculture and/or household work. Assets like refrigerators (69.7%) and televisions (79.5%) were used as socioeconomic markers.

Table 3 presents the results of weight-for-height and height-for-age assessments by sex when participants were ≤5 years old (2002–2003); 30.3% were overweight/obese (23.6% girls and 37.0% boys) and 72.1% were stunted. A high prevalence of malnutrition was observed in children; however, no statistically significant difference was found by sex. 

In the anthropometric assessment by sex in adulthood, men had a higher prevalence of overweight compared to women (43.5% and 27.6%, respectively), but no significant statistical differences were found. However, women had a higher prevalence of obesity compared to men (17.1% and 4.3%, respectively; *p* < 0.05). Women (53.9%) had abdominal obesity 1.8 times higher than men (30.4%). In total, 74.6% of the study participants presented risk of metabolic diseases when the data were analyzed by the WtHR (≥0.5 units); no significant statistical difference was found by sex. 

In the information about energy intake in 24 h, no significant difference was found by sex; however, their kilocalorie intake is above that recommended by the recommended dietary allowance (RDA) for adults aged 19 to 30 years (women and men). Regarding physical activity, women (64.5%) were more likely to be sedentary (<600 METs/week) than men (30.4%) (*p* < 0.05).

Table 4 shows the results of clinical, biochemical parameters and MetS diagnostic by sex. Elevated systolic blood pressure (SBP) (≥130 mmHg) was only present in men, with a prevalence of 17.4% (*p* < 0.05). The prevalence of elevated diastolic blood pressure (DBP) (≥85 mmHg) was 13% in men and 2.6% in women; no significant statistical difference was found.

Women had a higher prevalence of elevated glucose (18.2%) and high triglycerides (43.9%) than men (12.9% and 35.5%, respectively). Men had a higher prevalence of high cholesterol (29%), high LDL-c (6.5%) and low HDL-c (74.2%) compared to women (13.6%, 3% and 69.7%, respectively); however, no statistically significant differences were found. 

The prevalence of IR was higher in women (72.7%) than in men (48.4%) (*p* < 0.05), and men presented a higher prevalence of MetS compared to women (men 83.9% and women 78.8%) using the IDF criteria (*p* > 0.05). The overall prevalence of MetS was 19.6%.

The correlations between the independent variables are shown in Figure 3. Weight for height at childhood was significantly correlated with anthropometric parameters in adulthood: WC, WHtR and BMI. No statistically significant correlations were observed between height for age at childhood and cardiometabolic risk factors.

Binary logistic regression analysis of anthropometric assessment is presented in Table 5. In crude models, overweight/obesity in early childhood was associated with an increase of abdominal obesity (OR = 2.53, 95% CI: 1.05–6.07) and WHtR (OR = 3.84, 95% CI: 1.24–11.93) in adulthood. After adjusting for sex, age, ethnicity, total energy intake and physical activity, the association remained statistically significant for abdominal obesity (3.78, 95% CI: 1.55–9.24), WHtR (OR = 5.38, 95% CI: 1.60–18.10) and overweight/obesity by BMI (OR = 2.65, 95% CI: 1.09–6.45).

Table 6 shows the binary logistic regression analysis of the clinical and biochemical assessment. In the crude models, overweight/obesity in early childhood was associated with having high total cholesterol (OR = 3.95, 95% CI: 1.36–11.43) and MetS (OR = 3.45, 95% CI: 1.22–9.75) in adulthood. After adjusting for sex, age, ethnicity, total energy intake and physical activity, the association remained statistically significant for both indicators (OR = 6.43, 95% CI: 1.77–23.36; OR = 4.71, 95% CI: 1.49–14.90, respectively).

## 4. Discussion

Malnutrition in childhood affects health status at different stages of life [35,36,37,38]. The results of this study of adults living in highly marginalized communities in Chiapas, Mexico, carried out 18 years after a cohort study baseline, suggest that children under five years of age who had overweight/obesity are more likely to develop cardiometabolic risk factors in young adulthood.

Obesity has been associated with comorbidities that contribute to a high prevalence of disability, mortality and premature death in individuals, which is considered a public health problem [39]. In Mexico [15], 74.1% of the population over 20 years of age is overweight and obese; 74.5% and 72.6% in urban and rural areas, respectively, are overweight/obese. In the South Pacific region of Mexico, where the state of Chiapas is located, the prevalence of overweight/obesity in the population ≥ 20 years of age was slightly lower than the national average (72.2%).

In our study, a prevalence of overweight/obesity of 45.9% was obtained. In Chiapas, some studies [40,41] have reported a prevalence of overweight/obesity of 88% in women from Mayan communities older than 20 years, and 56.9% in indigenous and mestizo populations older than 24 years.

Additionally, we found a prevalence of abdominal obesity of 45.1%, and 74.6% cardiometabolic risk according to WHtR. Men had a higher prevalence of elevated SBP/DBP (17.4%/13%) compared to women. 

In Mexico, 81% of the adult population older than 20 years has abdominal obesity and 30.2% has arterial hypertension. Similar to our findings, the prevalence of hypertension is higher in men (10.1%) than in women (4.8%) from 20 to 29 years [39]. Our results on abnormal blood pressure prevalence are lower than those found in the literature [15,39]; this may be due to the age differences, since our study population is younger (21 ± 3 years).

With regards to biochemical indicators, we found that the prevalence of high total cholesterol (18.6%) and high triglycerides (41.2%) are very similar to those reported in previous studies in Mexico (18.7% high cholesterol and 42.5% high triglycerides). The prevalence of high LDL-c (4.1%) was lower than the country as a whole (16.1% high LDL-c), in contrast to the prevalence of low HDL-c (71.1%), which was higher in the study population than that of Mexico (30.8% low HDL-c) [13].

In our study population, the prevalence of abnormal blood glucose levels was 16.5%. The prevalence of diabetes in the Mexican population aged 20–29 years is lower (3.3%) than that found in this study [42]. A prevalence of IR of 64.9% was found, with a statistically significant difference by sex (women 72.7% and men 48.4%, *p* = 0.019). This may be due to the high prevalence of obesity presented by women in this study. Previous studies in Mexico found a lower prevalence of IR (12.1% and 15.3% for women and men, respectively) [43]. In another study conducted in a population of 42.36 ± 15.8 years in Veracruz, Mexico, a prevalence of 34.5% of IR was found [44].

IR has been extensively linked to chronic low-grade inflammation and the production of proinflammatory cytokines; it, just by itself, has been established as an independent risk factor for cardiovascular events, even in patients without diabetes [45].

The prevalence of MetS in the present study was 19.6%; women had a higher prevalence than men (21.2% and 16.1%, respectively), although this difference was not statistically significant. In a multicenter study carried out in Latin American countries (Mexico, Colombia, Brazil, Portugal and Argentina) [46], with apparently healthy volunteers under 28 years of age, the overall prevalence of MetS was 15.5% (23.1% in men and 12.1% in women), and for Mexico it was 26.6%, identified with the Harmonization Criteria for MetS proposed by Alberti et al. in 2009 [47]. The major prevalences for the main components of MetS were for abdominal obesity and low HDL levels. Murguía-Romero et al. [43] reported a prevalence of MetS of 13.4% in Mexicans aged 19.5 ± 1, and 71% had at least one of the five parameters of MetS.

In a meta-analysis, with an apparently healthy Mexican adult population, the prevalence of MetS was 54%, using the IDF criteria [48]. In Veracruz, Mexico, the MetS prevalence obtained by the WHO criteria was 18.5% [44]. On the other hand, a study among the Mexican indigenous population (47.9 ± 16.4 years of age) reported a prevalence of 50.3% based on the criteria of the National Heart, Lung and Blood Institute [49]. However, in Chiapas, women from Mayan communities under 35 years of age presented a prevalence of MetS of 53.4% through the IHC criteria [40].

MetS is a set of risk factors for developing diabetes and cardiovascular diseases [50,51], which are the first causes of mortality in Mexico and Chiapas [12,52]. Our study population had a prevalence of MetS different from that found in Latin America and in the different studies mentioned above. It should be noted that the criteria used for MetS and age ranges may explain the disparities between the results. Our data are consistent with those reported for Mexico according to age; more than half of the study sample presented at least one of the following cardiometabolic risk factors: overweight/obesity, abdominal obesity, low HDL-c and IR. Abdominal adiposity and IR constitute the pathophysiological basis of metabolic syndrome, diabetes and other chronic non-communicable diseases such as cardiovascular and renal diseases [53]. Vascular stiffness caused by IR and obesity has been significantly associated with damage to target organs, such as the heart, kidneys and brain [54]. In our study, weight-for-age Z-score in childhood was significantly and positively correlated with WC, WHtR and BMI, suggesting that higher weight in early childhood increases the likelihood of obesity in adulthood and increases the risk of developing cardiometabolic diseases. Several studies indicate that overweight children are more likely to be obese in adulthood [55,56]. Likewise, some studies report an association between childhood weight with cardiovascular risk factors and mortality in adulthood [57].

One of the most important results in this study was the association found between nutritional status during childhood and cardiovascular risk in adulthood. When analyzing the relationship between overweight/obesity presented in early childhood with anthropometric, clinical and biochemical indicators, we found statistically significant associations in some parameters of cardiovascular risk factors. Children under five years of age who had overweight/obesity presented a higher probability of having overweight/obesity (2.65 odds), abdominal obesity (3.78 odds), high WHtR (5.38 odds), high total cholesterol levels (6.43 odds) and MetS (4.71 odds) in adulthood.

In our population, according to the nutritional status in early childhood, 30.3% presented overweight/obesity and 72.1% had stunting. Such prevalences are above those reported at the national and regional level [15]. These results are significantly important since a previous meta-analysis concluded that overweight/obesity in childhood is moderately associated with an increased risk of morbidity in adulthood [58]. Children with obesity are more likely to have obesity in adulthood; 55% of children with obesity will continue to have obesity in adolescence, and about 80% of these adolescents will remain with this condition in adulthood [59]. In addition to this, childhood growth retardation can lead to reduced lean mass, since people with undernutrition tend to use protein reserves in the muscles, which can favor metabolic disorders in later stages of life [2].

Our findings agree with those of longitudinal studies conducted in Brazil and Guatemala [10,60,61] that show an association between childhood obesity and having obesity in adulthood. For other parameters, Callo Quinte et al. [60] found that overweight throughout life was associated with higher systolic and diastolic blood pressure, random abnormal glycemia and lower HDL-c levels, but no association was found with high triglycerides and LDL-c levels. Similarly, evidence suggests that a higher BMI in childhood is an independent risk factor for increased total cholesterol, LDL-c and triglycerides, as well as low HDL-c concentrations and high blood pressure [53]. Nonetheless, Kroker-Lobos et al. [10] did not find associations between childhood obesity and hypertension and MetS in adulthood. 

Evidence suggests that the origins of dyslipidemia and cardiovascular diseases can be determined in childhood, along with obesity [37,53]. Obesity in childhood accelerates the process of atherogenesis, leading to the development of atherosclerosis, causing changes in blood vessels due to the appearance of fatty streaks, and atherosclerotic wall lesions are in direct connection with childhood obesity [59]. 

Our results showed a non-significant trend of presenting abnormal levels regarding blood pressure, glucose, triglycerides and LDL-c, and low HDL-c, as well as insulin resistance. Some studies have shown the relationship between childhood obesity and increased total cholesterol, LDL-c, triglycerides and low HDL-c concentrations in adulthood [59,62]. 

One possible explanation for these findings is that our study population has its own and different characteristics. Many participants are indigenous (35.2%), living in communities facing significant challenges like high degrees of marginalization, extreme poverty, migration and social exclusion [63,64,65]. Additionally, their diet [21,66] and a sedentary lifestyle (51.6%) may also contribute to these outcomes. Some studies have suggested that a genetic background plays an important role in the susceptibility to these diseases, in addition to a change in lifestyle that has been reflected in the last 18 years [21,40,63,67].

In this study, total energy intake was used as a covariate; however, we consider it is important to further analyze the quality of the diet related to cardiometabolic risk factors in this population. These results would provide information on the nutritional transition that has occurred in Chiapas and its impact on health.

Among the limitations of our study, the sample size was limited due to the loss rate of the cohort sample for this follow up study (70%): migration (62.1%), refusal to participate (36.5%) and mortality (1.4%). However, there are no reasons to think that the characteristics of the sample are different to non-participants. Among the strengths of this study, there are no studies apart from those reported by the ECOSUR Research team on this population [17,18,21,22,23,66,67,68]. Moreover, the longitudinal design allowed us to explore the changes in nutritional and health status of this population after 18 years. In addition, the association between malnutrition and cardiovascular risk factors has not been reported in previous studies for the state of Chiapas.

## 5. Conclusions

In summary, our results indicate that overweight/obesity in early childhood was associated with an adverse cardiometabolic profile in adult populations living in marginalized communities in southern Mexico. The evidence provided by this study suggests the importance of not only early intervention during childhood to reduce obesity but also throughout the life course to reduce the risk of developing cardiometabolic diseases that contribute to the disease burden in marginalized communities in Chiapas, Mexico.

## Figures and Tables

**Figure 1 nutrients-17-00254-f001:**
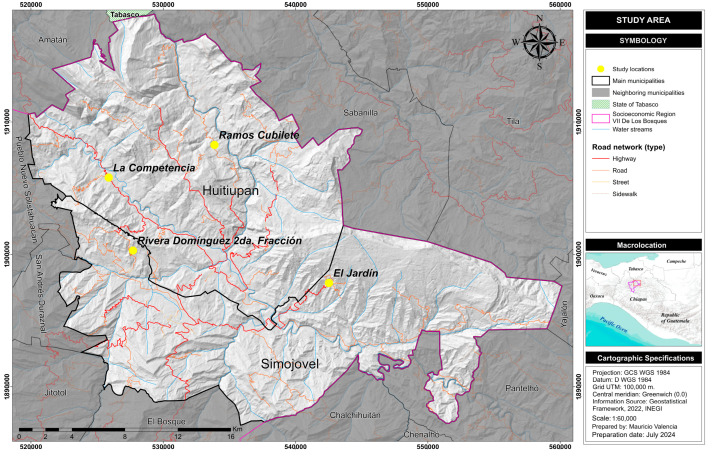
Study area: De Los Bosques region, Chiapas, Mexico.

**Figure 2 nutrients-17-00254-f002:**
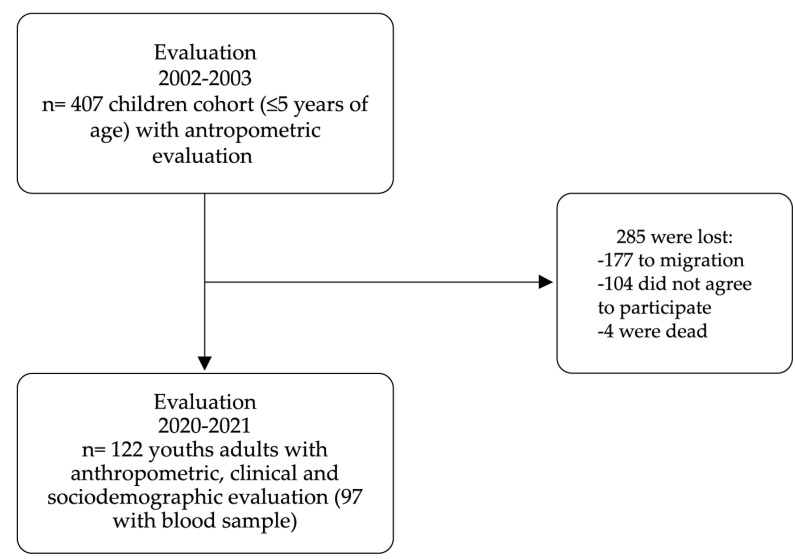
Cohort flowchart.

**Figure 3 nutrients-17-00254-f003:**
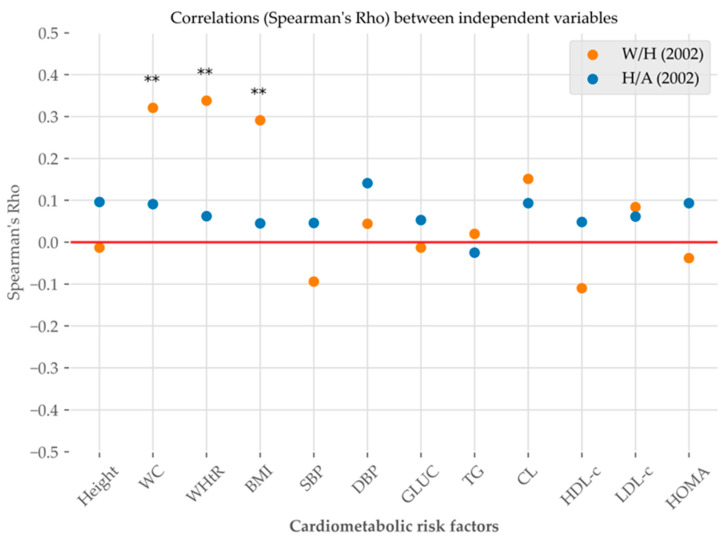
Comparison of Spearman’s correlation coefficient between weight-for-height index (W/H) (orange) and height-for-age index (H/A) (blue), in childhood (2002), and cardiometabolic risk factors: height, waist circumference (WC), waist to height ratio (WHtR), body mass index (BMI), systolic blood pressure (SBP), diastolic blood pressure (DBP), glucose (GLUC), triglycerides (TG), cholesterol (CL), high-density lipoprotein cholesterol (HDL-c), low-density lipoprotein cholesterol (LDL-c) and homeostasis model assessment (HOMA) in adulthood (2021). ** The correlation is significant at the 0.01 level (two-sided).

**Table 1 nutrients-17-00254-t001:** Cut-off points for biochemical parameters.

Biochemical Parameters	Cut-Off Points
Glucose (mg/dL) [28]	Normal: <100 High: ≥100
Total cholesterol (mg/dL) [28]	Normal: <200 High: ≥200
Triglycerides (mg/dL) [28]	Normal: <150 High: ≥150
HDL-c (mg/dL) [28]	Normal: >50 for woman and >40 for men Low: ≤50 for women and ≤40 for men
LDL-c (mg/dL) [31]	Normal: <130 High: ≥130
HOMA-IR [30]	Normal: <2.6 IR: ≥2.6

Abbreviations: HDL-c, high-density lipoprotein cholesterol; LDL-c, low-density lipoprotein cholesterol; HOMA-IR, homeostatic model assessment insulin resistance; IR, insulin resistance.

**Table 2 nutrients-17-00254-t002:** Sociodemographic characteristics of young adults from Chiapas, Mexico, by sex ^1^.

Variables	Women		Men		Total		
	n	Median (IQR) or %	n	Median (IQR) or %	n	Median (IQR) or %	*p*-Value
Age (years)							0.870 ^2^
18–25	76	21(3)	46	21(2)	122	21(3)	
Marital status							0.401 ^3^
With partner	39	51.3	20	43.5	59	48.4	
Single	37	48.7	26	56.5	63	51.6	
Total	76	100	46	100	122	100	
Localities							
La Competencia	33	43.4	15	32.6	48	39.3	0.237 ^4^
Ramos Cubilete	26	34.2	19	41.3	45	36.9	0.431 ^4^
Rivera Domínguez	13	17.1	10	21.7	23	18.9	0.529 ^4^
El Jardín	4	5.3	2	4.3	6	4.9	0.805 ^4^
Total	76	100	46	100	122	100	
Language (ethnicity)							0.485 ^3^
Spanish	51	67.1	28	60.9	79	64.8	
Indigenous	25	32.9	18	39.1	43	35.2	
Total	76	100	46	100	122	100	
Years of schooling							0.202 ^3^
≥6 years	59	77.6	40	87.0	99	81.1	
<6 years	17	22.4	6	13.0	23	18.9	
Total	76	100	46	100	122	100	
Literacy							0.402 ^3^
Read and write	71	93.4	41	89.1	112	91.8	
Illiterate	5	6.6	5	10.9	10	8.2	
Total	76	100	46	100	122	100	
Occupation							0.086 ^3^
Farmer	63	82.9	32	69.6	95	77.9	
Another job	13	17.1	14	30.4	27	22.1	
Total	76	100	46	100	122	100	
Household conditions							
Fridge							0.215 ^3^
Yes	56	73.7	29	63.0	85	69.7	
No	20	26.3	17	37.0	37	30.3	
Total	76	100	46	100	122	100	
Television							0.098 ^3^
Yes	64	84.2	33	71.7	97	79.5	
No	12	15.8	13	28.3	25	20.5	
Total	76	100	46	100	122	100	

^1^ The data for the quantitative variables are represented in medians and Interquartile range (IQR) (difference between the third quartile and first quartile) and the data for the qualitative variables are represented in percentages. ^2^ Mann–Whitney U test. ^3^ Pearson’s chi square. ^4^ Z-test for comparison of proportions.

**Table 3 nutrients-17-00254-t003:** Anthropometric parameters of the study population by sex ^1^.

Variables	Women		Men		Total		
	n	Mean ± SD or %	n	Mean ± SD or %	n	Mean ± SD or %	*p*-Value
*Nutritional evaluation in early childhood (≤5 years-2002)*							
Weight for height (z-score)	76	0.2 ± 1.6	46	−0.2 ± 1.6	122	0.4 ± 1.3	0.200 ^2^
Normal	56	73.7	29	63.0	85	69.7	0.219 ^3^
Overweight/obesity (≥+1 SD)	20	26.3	17	37.0	37	30.3	
Total	76	100	46	100	122	100	
Height for age (z-score)	76	±1.8 (1.9) ^4^	46	−2.0 (2.7) ^4^	122	−1.8 (2.0) ^4^	0.501 ^5^
Normal	22	28.9	12	26.1	34	27.9	0.733 ^3^
Stunting (≤−2 SD)	54	71.1	34	73.9	88	72.1	
Total	76	100	46	100	122	100	
*Nutritional evaluation in adults (18–25 years)*							
Height (m)	76	1.51 (0.05) ^4^	46	1.63 (0.08) ^4^	122	1.53 (0.12) ^4^	0.550 ^5^
Weight (kg)	76	57.6 ± 11.6	46	65.2 ± 11.8	122	60 ± 11.7	0.200 ^2^
BMI (kg/m^2^)	76	25.8 ± 4.7	46	24.6 ± 3.9	122	25.1 ± 4.4	0.200 ^2^
Normal (18.5–24.9)	42	55.3	24	52.2	66	54.1	0.739 ^6^
Overweight (25–29.9)	21	27.6	20	43.5	41	33.6	0.072 ^6^
Obesity (≥30)	13	17.1	2	4.3	15	12.3	0.037 ^6^
Total	76	100	46	100	122	100	
Waist circumference (cm)	76	83.1 ± 12.7	46	85.6 ± 8.5	122	83.9 ± 11.2	0.082 ^2^
Normal (women < 80 and men < 90)	35	46.1	32	69.6	67	54.9	0.011 ^3^
Abdominal obesity (women ≥ 80 and men ≥ 90)	41	53.9	14	30.4	55	45.1	
Total	76	100	46	100	122	100	
Waist to height ratio (units)	76	0.5 (0.1) ^4^	46	0.5 (0.1) ^4^	122	0.5 (0.1) ^4^	0.090 ^5^
Normal (<0.5)	20	26.3	11	23.9	31	25.4	0.768 ^3^
High (≥0.5)	56	73.7	35	76.1	91	74.6	
Total	76	100	46	100	122	100	
Physical activity	76	222.9 (1177.1) ^4^	46	2417 (2854.3) ^4^	122	480 (2291.4) ^4^	≤0.001 ^5^
Active (>600 METs/week)	27	35.5	32	69.6	59	48.4	≤0.001 ^3^
Sedentary (<600 METs/week)	49	64.5	14	30.4	63	51.6	
Total	76	100	46	100	122	100	
Energy intake in 24 h (kcal/day)	76	2096.7 (1194.2) ^4^	46	2117.8 (844.4) ^4^	122	2101.1 (971.7) ^4^	0.907 ^5^

^1^ The data for the quantitative variables are represented in mean and standard deviation (SD) and the data for the qualitative variables are represented in percentages; ^2^ Kolmogorov–Smirnov; ^3^ Pearson’s chi square; ^4^ Medians and interquartile range (IQR) (difference between the third quartile and first quartile) show; ^5^ Mann–Whitney U test; ^6^ Z-test for comparison of proportions. Abbreviations: BMI, body mass index; MET, metabolic equivalent of task.

**Table 4 nutrients-17-00254-t004:** Clinical and biochemical parameters of the study population by sex.

Variables	Women		Men		Total		
	n	%	n	%	n	%	*p*-Value
*Clinical*							
SBP							0.000 ^1^
Normal (<130)	76	100	38	82.6	114	93.4	
High (≥130)	0	0	8	17.4	8	6.6	
Total	76	100	46	100	122	100	
DBP (mm Hg)							0.052 ^1^
Normal (<85)	74	97.4	40	87.0	114	93.4	
High (≥85)	2	2.6	6	13.0	8	6.6	
Total	76	100	46	100	122	100	
*Biochemical*							
Glucose							0.574 ^1^
Normal (<100 mg/dL)	54	81.8	27	87.1	81	83.5	
High (≥100 mg/dL)	12	18.2	4	12.9	16	16.5	
Total	66	100	31	100	97	100	
Triglycerides							0.510 ^2^
Normal (<150 mg/dL)	37	56.1	20	64.5	57	58.8	
High (≥150 mg/dL)	29	43.9	11	35.5	40	41.2	
Total	66	100	31	100	97	100	
Cholesterol							0.069 ^2^
Normal (<200 mg/dL)	57	86.4	22	71.0	79	81.4	
High (≥200 mg/dL)	9	13.6	9	29.0	18	18.6	
Total	66	100	31	100	97	100	
LDL-c							0.591 ^1^
Normal (<130 mg/dL)	64	97.0	29	93.5	93	95.9	
High (≥130 mg/dL)	2	3.0	2	6.5	4	4.1	
Total	66	100	31	100	97	100	
HDL-c							0.649 ^2^
Normal (>50 mg/dL women and >40 mg/dL men)	20	30.3	8	25.8	28	28.9	
Low (≤50 mg/dL women and ≤40 mg/dL men)	46	69.7	23	74.2	69	71.1	
Total	66	100	31	100	97	100	
HOMA –IR (units)							0.019 ^2^
Normal < 2.6	18	27.3	16	51.6	34	35.1	
IR ≥ 2.6	48	72.7	15	48.4	63	64.9	
Total	66	100	31	100	97	100	
MetS (IDF)							0.784 ^2^
No MetS	52	78.8	26	83.9	78	80.4	
With MetS	14	21.2	5	16.1	19	19.6	
Total	66	100	31	100	97	100	

^1^ Fisher’s exact test; ^2^ Pearson’s chi square. Abbreviations: DBP, diastolic blood pressure; HDL-c, high-density lipoprotein cholesterol; HOMA-IR, homeostatic model assessment insulin resistance; IDF, International Diabetes Federation; IR, insulin resistance; LDL-c, low-density lipoprotein cholesterol; MetS, metabolic syndrome; SBP, systolic blood pressure.

**Table 5 nutrients-17-00254-t005:** Association between weight for height in childhood and cardiometabolic risk factors (anthropometric) in the study population.

Variables	Anthropometric and Clinical Evaluation in Young Adults			
Indicators in Infants ≤ 5 Years	% (n)	% (n)	Model I OR (95% CI) ^1^	Model II OR (95% CI) ^2^
*Body mass index*	Normal	Overweight/obesity		
Normal weight for height (<2 SD)	75.8 (50)	62.5 (35)	1 (Ref.)	1 (Ref.)
Overweight/obesity (≥+ 1 SD)	24.2 (16)	37.5 (21)	1.87 (0.86–4.09)	2.65 (1.09–6.45) ^4^
*Waist circumference*	Normal	Abdominal obesity		
Normal weight for height (<2 SD)	79.1 (53)	58.2 (32)	1 (Ref.)	1 (Ref.)
Overweight/obesity (≥+ 1 SD)	20.9 (14)	41.8 (23)	2.53 (1.05–6.07) ^3^	3.78 (1.55–9.24) ^4^
*Waist to height ratio*	Normal	≥0.5 units		
Normal weight for height (<2 SD)	87.1 (27)	63.7 (58)	1 (Ref.)	1 (Ref.)
Overweight/obesity (≥+ 1 SD)	12.9 (4)	36.3 (33)	3.84 (1.24–11.93) ^3^	5.38 (1.60–18.10) ^4^

^1^ Crude model; ^2^ Adjusted for sex (males/females), age (years), language (Spanish or Indigenous), energy consumed in 24 h (kcal/day) and physical activity (active/sedentary); ^3^ *p* < 0.05; ^4^ *p* < 0.001.

**Table 6 nutrients-17-00254-t006:** Association of weight for height in infants and cardiometabolic risk factors (clinical and biochemical) and MetS in the study population.

Variables	Clinical and Biochemical Evaluation			
Indicators in Infants ≤ 5 Years	% (n)	% (n)	Model I OR (95% CI) ^1^	Model II OR (95% CI) ^2^
*Systolic blood pressure*	Normal	High		
Normal weight for height (<2 SD)	69.3 (79)	75 (6)	1 (Ref.)	1 (Ref.)
Overweight/obesity (≥+ 1 SD)	30.7 (35)	25 (29)	1.33 (0.26–6.91)	1.85 (0.23–14.71)
*Diastolic blood pressure*	Normal	High		
Normal weight for height (<2 SD)	71.1 (81)	50 (4)	1 (Ref.)	1 (Ref.)
Overweight/obesity (≥+ 1 SD)	29.9 (33)	50 (4)	2.45 (0.58–10.40)	2.29 (0.52–10.02)
*Glucose*	Normal	High		
Normal weight for height (<2 SD)	73.7 (42)	65 (26)	1 (Ref.)	1 (Ref.)
Overweight/obesity (≥+ 1 SD)	26.3 (15)	35 (14)	1.51 (0.63– 3.63)	1.84 (0.72–4.69)
*Total Cholesterol*	Normal	High		
Normal weight for height (<2 SD)	75.9 (60)	44.4 (8)	1 (Ref.)	1 (Ref.)
Overweight/obesity (≥+ 1 SD)	24.1 (19)	55.6 (10)	3.95(1.36–11.43) ^3^	6.43 (1.77–23.36) ^4^
*HDL-c*	Normal	Low		
Normal weight for height (<2 SD)	78.6 (22)	66.7 (46)	1 (Ref.)	1 (Ref.)
Overweight/obesity (≥+ 1 SD)	21.4 (6)	33.3 (23)	1.83 (0.65–5.15)	2.09 (0.70–6.20)
*LDL-c*	Normal	High		
Normal weight for height (<2 SD)	71 (66)	50 (2)	1 (Ref.)	1 (Ref.)
Overweight/obesity (≥+ 1 SD)	29 (27)	50 (2)	2.44 (0.33–18.25)	4.98 (0.54–45.69)
*HOMA-IR*	Normal	IR		
Normal weight for height (<2 SD)	64.7 (22)	73 (46)	1 (Ref.)	1 (Ref.)
Overweight/obesity (≥+ 1 SD)	35.3 (12)	27 (17)	1.48 (0.60–3.62)	1.09 (0.41–2.87)
*MetS (IDF)*	Normal	MetS		
Normal weight for height (<2 SD)	75.6 (59)	47.4 (9)	1 (Ref.)	1 (Ref.)
Overweight/obesity (≥+ 1 SD)	24.4 (19)	52.6 (10)	3.45 (1.22–9.75) ^3^	4.71(1.49–14.90) ^4^

^1^ Crude model; ^2^ Adjusted for sex (males/females), age (years), language (Spanish or Indigenous), energy consumed in 24 h (kcal/day) and physical activity (active/sedentary); ^3^ *p* ≤ 0.05; ^4^ *p* ≤ 0.001. Abbreviations: HDL-c, high-density lipoprotein cholesterol; HOMA-IR, homeostatic model assessment insulin resistance; IDF, International Diabetes Federation; IR, insulin resistance; MetS, metabolic syndrome; LDL-c, low-density lipoprotein cholesterol.

## Data Availability

The data presented in this study are available on request from the corresponding authors.

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
