# Peer review of "Impact of Early Childhood Malnutrition on Cardiometabolic Risk Factors in Young Adults from Marginalized Areas of Chiapas, Mexico"

_nutrients, 2025, doi:10.3390/nu17020254_

Round 1

Reviewer 1 Report

Comments and Suggestions for Authors

Dear authors, The subject chosen to be analyzed in this study is a very important and topical one not only in Mexico, but in the whole world, because malnutrition in children is a main cause of morbidity and mortality worldwide.  Another main cause of morbidity and mortality in all age groups, regardless of the socioeconomic status of the respective country or region, is obesity and metabolic syndrome. And there are more and more studies that demonstrate the close connection between the two morbid conditions. 

I believe that the study is an important one, and the manuscript should be considered for publication after some corrections.

Introduction: A short phrase with reference to the world situation (not only the one in Mexico) regarding the impact of malnutrition and obesity in children on adult health would be necessary.

Material and methods: The second sentence (line 80-82) in this section is not necessary.

-Figure 2 is incomplete

- Better insert  biochemical parameters (line 156-166) in a table

Results: Childhood overweight or obesity was associated with an increased risk of adult overweight or obesity, a high waist circumference, a high waist-to-height ratio, raised total cholesterol, and Metabolic Syndrome. Your results are in accordance with other results of international studies.

Discussion: -Line 330 and 336: please define RI and IR

Conclusion is well written. 

Reviewer 2 Report

Comments and Suggestions for Authors

This study includes a large amount of data and several variables which compensates for the small size of the sample. The design and results are interesting, although tracking of child overweight/obesity into adulthood is nothing new. We have a few queries and suggestions.

1. The title is somewhat misleading as it 'malnutrition' in childhood usually refers to undernutrition. It is suggested to use overweight/obesity instead of malnutrition.

2.  Something is unclear about the sample.  What do the second and third assessment mean? More detail is also needed regarding the 'intentional' (meaning purposive?) sample of the 407 children.

3. The communities are described as 'marginalized': please explain in the introduction why they are so.

4.  Why are medians used systematically for quantitative data? All variables were skewed?

5.  Figure 1 is not clear and Figure 2 needs to be re-designed.

6.  References on the tracking of child overweight/obesity into adulthood and the impact of child obesity on cardiometabolic risk in adulthood are incomplete. At least these three should be referred to: Singh AS et al 2008; Freedman DS et al 2001 (Bogalus Study); and Park MH et al (2012).

7. Dietary data are limited to energy intake whilst diet quality could have been assessed. This is likely saved for another paper, in which case it should be mentioned. 

8. Higher height-for-age Z-score in childhood was significantly and positively correlated with BMI and abdominal obesity markers in adulthood: this should be discussed. 

9.  Line 160: Change 'high' HDLC for 'low'.

Round 2

Reviewer 1 Report

Comments and Suggestions for Authors

Dear authors,

Thank you for your  answers to my questions.

Your manuscript is improved and can be published.

Best regards,